# Effects of the Density of Invasive *Lantana camara* Plants on the Biodiversity of Large and Small Mammals in the Groenkloof Nature Reserve (GNR) in South Africa

**DOI:** 10.3390/biology12020296

**Published:** 2023-02-13

**Authors:** Tlou D. Raphela, Kevin J. Duffy

**Affiliations:** 1Institute of System Science, Durban University of Technology, 4-43 M L Sultan Road, Durban 4000, South Africa; 2Disaster Management Training and Education Centre for Africa, University of the Free State, Bloemfontein 9301, South Africa

**Keywords:** weeds, invasion, biodiversity, protected areas

## Abstract

**Simple Summary:**

Weeds have been extensively reported to cause numerous disturbances in ecosystems worldwide. However, the impacts of these weeds on biodiversity, especially for mammals, has received little attention in South Africa. Therefore, this study explored the impact of one specific weed, *Lantana camara*, on the mammals of an urban Game Reserve in South Africa. Small mammal weights were varied by treatment type and the degree of invasion. Moreover, the weight of these small mammals varied by season across different treatments. This finding implies that the weight of the small mammals in the study was seasonally affected by the quantity of weeds. For large mammals, the distance of treatments (where large mammal tracks were sampled) from a water source was not a significant predictor of species richness, but density of *L. camara* was a significant predictor of large mammal species richness. This study concludes that the *L. camara* weed influences mammals of the Groenkloof Nature Reserve (GNR) negatively.

**Abstract:**

Multi-scale approaches have been used to determine scales at which mammal species are responding to habitat destruction due to invasion, but the impacts of weeds on mammals have not been extensively studied, especially in Africa. Inside the Groenkloof Nature Reserve (GNR), we assessed how mammals are affected by an invasive weed *Lantana camara*. A series of models were applied to determine the differences in species abundance as well as richness, separated for large and small mammals. When diversity indices were used, an Analysis of Variance (ANOVA) revealed no statistically significant difference between treatments (F_5_ = 0.233, *p* = 0.945) for large mammals. The results of a Generalised Linear Mixed Model (GLMM) showed that vegetation type (Wald χ^2^_2_ = 120.156; *p* < 0.01) and foraging guilds (Wald χ^2^_3_ = 76.771; *p* < 0.01) were significant predictors of large mammal species richness. However, for small mammals, the results of a GLMM showed that only treatment type (Wald χ^2^_5_ = 10.62; *p* = 0.050) was a significant predictor of the number of small mammals trapped. In addition, the ANOVA revealed statistically significant differences in species diversity between treatments (F_5_ = 0.934; *p* < 0.001) and by season (F_1_ = 9.122 *p* = 0.003) for small mammals. The presence of *L. camara* coupled with other predictors was associated with differences in large mammal abundances and diversity, and differences in how these large mammals were distributed across the landscape. Furthermore, the highest species diversity was found in the spring for small mammals. Therefore, for all the mammals studied, the presence of *L. camara* negatively affected species abundance, richness, and diversity, as well as how these species were distributed across the invaded and cleared areas.

## 1. Introduction

Globally, invasive species have severe negative impacts on biodiversity [1,2,3]. In Australia, van Wilgen et al. [4] identified 1321 native plants and 158 native animal species that are threatened by the invasive weed *Lantana camara*, and that 275 of the native plants and 24 native animal species required immediate protection from *L. camara* invasions. This could be because *L. camara*, like most invasive shrubs, affects the grass layer [5,6]. The impacts of invasive weeds on terrestrial mammals have received little attention, even though these mammals are reported to promote ecotourism and improve the economy of the communities surrounding protected areas [7].

### Problem

Worldwide habitat loss due to invasive species has been reported by numerous studies [8,9]. Stuart et al. [8] reported that these losses threaten 48% of rapidly declining animal species and are driving species, especially mammals, to extinction. The lack of conservation solutions for these invasive plant-species-induced declines could mean that hundreds of mammals worldwide now face extinction, which will be a serious and holistic problem for biodiversity. As habitats are altered, an untold number of mammals are disappearing before they have been recognised, much less studied [9]. This is especially true for protected areas in most urban setups of developing countries. For example, less mammal research has been conducted in the study area, most likely because the Groenkloof Nature Reserve (GNR) is very commercialised, and some research methodologies will not work due to aspects such as noise pollution [2]. This leaves the effects of invasive species in this type of setup unstudied. Ceballos et al. [10] documented the population extinctions in 177 mammals and found that the rate of population loss in terrestrial mammals due to invasive weeds was extremely high between 1900 and 2015, which is a serious problem in itself. Furthermore, Amin et al. [11] identified invasive weeds as a threat to 212 mammals in Nepal. On oceanic islands, habitat loss due to introduced invasive plant species has driven the extinction of mammals [12]. In South Africa, invasive alien plants have also been recognised as a threat to native biodiversity for more than two decades [1].

Ewel et al. [13] identified the negative impacts of invasive species on the shelter and protection of mammals, and Ehrenfeld [14] reported that the thorny and bushy habitat of Japanese barberry (*Berberis thunbergii*) makes it an ineffective shelter for mammals. However, less attention has been paid to other weeds in the grassland’s biomes, especially in South Africa [15], and the importance of the present study is an effort to rectify this situation.

The *L. camara* leaves are reported to be unpalatable or even toxic to mammalian herbivores due to the presence of secondary plant metabolites, mainly triterpene acids. In addition, *L. camara* alters the quality of the habitat by potentially reducing the diversity and abundance of forage species available to mammals [16].

The present study is focused on the effects of *L.* camara on mammals of the GNR. For large mammals, the studies referenced above provide the bases of three hypotheses: (1) Large mammalian species richness will be affected by vegetation type but this will be dependent on foraging guild [17]; (2) *L. camara* will affect large mammalian species richness based on the distance of the treatments from a water source, with the expectation of higher species richness in treatments nearer to a water source [18]; (3) Large mammal species richness will be lower in areas with higher *L. camara* density compared to areas with lower *L. camara* density [19].

The impacts of most invasive species on small terrestrial mammals have not been extensively quantified. Invasion biology focuses more on large flagship mammals for ecotourism purposes [20]. A literature scan has revealed only one study in England that explored the impacts of the invasive *Rhoddendron ponticum* shrub on wood mice (*Apodemus sylvaticus*). The lack of attention paid to the effects of biological invasion on small mammals [21,22] is surprising because these mammals are important in the ecosystem. Thus, case-specific studies are needed from different protected areas, with different invasion and clearing times, in order to enable the generation of a better understanding of biological invasion on a global level [23]. Ceradini and Chalfoun [24] reported a decrease in the occupancy of two rodent species due to invasive cheatgrass in North America, indicating modification in species composition. Similarly, Haack et al. [25] also reported that invasive *L. camara* influences small mammal assemblages because of its canopy and thickets. The invasive *Lonicera maackii* has been shown to change rodent foraging behaviour and decrease foraging activity because of the thicker foliage typical of invasive weeds [26]. In addition, weed thickets can cover native seeds preferred by granivorous rodents [27].

The canopies and thickets of *L. camara* can shift the community composition of small mammals adapted to warmer habitats and affect their assemblage. In this study, rodents were captured to assess the impacts of *L. camara* on the terrestrial small mammals and their assemblage inside the GNR with the objective of quantifying the impact of *L. camara* on these terrestrial small mammals inside the reserve. The study made two predictions as follows: (1) *L. camara* shrubs will have a negative effect on rodent species assemblage, and (2) the biggest rodents in terms of weight will be found in the control areas compared to smaller rodents and in the spring season. This last prediction is based on reports that rodents are more territorial in the spring season compared to other seasons [28,29].

## 2. Materials and Methods

### 2.1. Study Area

This study was conducted from October 2019 to February 2021 inside the Groenkloof Nature Reserve (GNR) in South Africa. The GNR (25°47′23″ S–28°11′46.1″ E) is 600 ha in extent [4]. The GNR has a diverse mammal fauna [30], and is mainly dominated by a variety of grasses, where animals spend most of their time grazing [31]. However, some areas inside the reserve have been converted into shrubland by *L. camara* invasion, limiting the growth of the grass that used to dominate. While grass still dominates overall, there are areas where *L. camara* dominates with higher densities compared to other areas.

In the early 1990s, Eckhardt et al. [32] recognized three vegetation types inside the GNR: mixed bushveld, highveld grassland, and rocky highveld grassland. Currently, there is an abundance of *L. camara* weed inside the reserve, as shown by the GNR invasive species abundance map, and this is a concern for the management of the GNR. However, grassland comprises 55% of the reserve [33]. There are bush-dominated areas inside the GNR, but the extent of these has not been documented.

The very patchy information regarding small mammal populations and diversity in the GNR is a severe constraint on proper management planning to assist in its biodiversity conservation [34]. The most common weeds in the GNR are pompom (*Eupatorium macrocephalum*) and common lantana (*L. camara*). *L. camara* is selected here because this weed is reported as amongst the most notorious and problematic of all invasive plants globally, and is ranked amongst the world’s ten worst weeds [12]. Compared to *E. macrocephalum, L. camara* is common in the study area and its threats and effects are undeniably recognizable in the park. Information about the environmental conditions of the study area (Table 1) was obtained from the GNR database.

### 2.2. Site Selection

To assess how *L. camara* affected large mammals inside the GNR, two sites with differing vegetation types were identified inside the reserve. Using historical maps (provided by GNR management), the study sampled two distinct sites threatened by *L. camara*, each with a different dominant vegetation type: a grass-dominated site and a bush-dominated site. The sites selected consisted of at least ~60% of the dominant vegetation, even though *L. camara* was also present in both sites. These sites were strategically selected so that some *L. camara* control work was conducted on both sites. Using the GNR *L. camara* clearing map, six treatment tracking plots were constructed within each site. In total, 12 five meters long and two meters wide treatment tracking plots were created for this study across the two sites. Large-scale control of *L. camara* was underway during the study because *L. camara* is the most problematic weed inside the reserve. During our study, the weed control team of the reserve was already removing large swaths of this weed using the cut-stump method of Rovero et al. [35]. The research team therefore worked with the *L. camara* control team to select the twelve treatment tracking plots for large mammal identification.

### 2.3. Treatment Design

To assess how *L. camara* affected mammals inside the GNR, the study analysed historical *L. camara* invasion and clearing maps (provided by GNR management) to identify and determine the study sites. Six treatment areas were selected from the maps that were equidistant, approximately 1 km from each other, after determining the duration of *L. camara* invasion and cleared periods. In total, twelve treatment tracking plots were selected across the different vegetation types, with six treatment plots in each vegetation type, grass-dominated and bush-dominated. These represented (1) plots that had been invaded by *L. camara* approximately 20 years (*ca*. 20 yrs) before the study commenced; (2) plots invaded for approximately for 10 years (*ca*. 10 yrs); (3) those invaded for approximately 2 years (*ca*. 2 yrs); (4) plots that were cleared between 3 and 5 years (*cl.* 3–5 yrs) from the start of the study; (5) plots that had been cleared in less than 2 years (*cl.* < 2 yrs); (6) plots that had never been invaded (the control plots) across each vegetation type. Differing clearing times, *cl.*, in this study is the number of years since *L. camara* was removed, and ca. refers to the period since an invasion. Because *L. camara* is very invasive, it is assumed that treatments *ca*. 2 yrs to *ca*. 20 yrs represent increasing *L. camara* and similarly for *cl.* < 2 yrs to *cl.* 3–5 yrs. The removal of every *L. camara* weed to create treatment tracking plots was conducted with the reserves’ *L. camara* clearing team for record keeping as management of this weed is continuous inside the GNR. Removal of all vegetation to create tracking plots was approved by Reserve Management for the City of Tshwane (Protocol number: GNR 2019/20). In addition, treatments were in areas with minimal active management activities to eliminate human disturbances.

### 2.4. Mammal Sampling

#### 2.4.1. Large Mammals

Data collected for this study (*L. camara* density and large mammal tracks) was demarcated to 100 m^2^ region for each treatment type. Similar studies by Dumalisile and Somers [15] and Rovero et al. [35] used a 100 m^2^ study area for large mammal sampling using tracks in two separate conservation areas. Throughout the present study, treatment tracking plots were kept relatively small for large mammals at 5 × 2 m to avoid clearing a larger area and interfering with the predator/prey hideouts. Esparton et al. [36] sampled large mammals on 0.5 × 0.5 m plots in a forest in Brazil. To track large mammal, 12 treatment tracking plots across the two vegetations type were demarcated in the GNR. The size of these plots was 2 × 5 m^2^.

The study used track counts to sample large mammals inside the GNR. All tracking plots were examined for tracks every second day for four hours from 06h00 to 08h00, and again from 16h00 to 18h00, throughout the study. After recording the tracks, the research team raked and smoothed the plots after adding smooth soil from the surrounding areas. Track counts are used across many different geographic areas to monitor the abundance of mammal species in most mammal-tracking studies [37]. To account for inter-observer variabilities, the double-observer protocol was adopted for this study [38]. The exercise was conducted twice, whereby two trained observers conducted the survey simultaneously [39].

Identifying tracks is a question of practice and experience, animal species seen often develop familiarity and confidence in identification [3]. Tracks of antelopes, such as *Tragelaphus strepsiceros*, *Hippotragus niger*, *Connochaetes taurinus*, and *Aepyceros melampus*, can be confusing. However, the size and the curves of their prints make them distinguishable from each other. For instance, *C. taurinus* tracks are bigger at the back and come to a point at the front that faces outwards, compared to the similarly shaped but smaller *Tragelaphus strepsiceros* tracks, which have sharper front points that face inside. *Tragelaphus strepsiceros*, *Hippotragus niger*, and *Aepyceros melampus* have somewhat similar tracks but the shape of their tracks at the back are different [39].

Two trained observers with digital cameras walked around the plot perimeters to identify and photograph any evidence of large mammals inside the plots and to record any evidence such as droppings and the number of tracks identified during each observation [39]. Distance of tracking plots from the water source were recorded using a Global Positioning System (GPS) and *L. camara* density was estimated inside the 100 m^2^ data collection area across all treatments by counting *L. camara* stems [40].

#### 2.4.2. Small Mammals

To assess how *L. camara* affected small mammals inside the GNR, the study analysed historical *L. camara* invasion and clearing maps (provided by GNR management) to identify and determine the study sites. Five treatment areas were selected from the maps that were equidistant, approximately 1 km from each other, after determining the duration of *L. camara* invasion and cleared periods: treatments that had been (1) recently invaded by *L. camara* (approximately *ca*. 2 years before the study commenced); (2) invaded for approximately 10 and 20 years before the study commenced; (3) cleared for less than 2 years; (4) cleared for between 3 and 5 years; and (5) ones never invaded by *L. camara*. Trapping quadrats of 100 m^2^ were created using white PVC poles demarcated in each treatment to sample small mammals [3].

A capture, mark, identify, and release protocol technique was used to sample small mammals [41,42,43] in each of the 12 different treatment areas. Trapping was conducted monthly from November 2019 to February 2020. Traps were set in the quadrats in each treatment ensuing a balanced design. A total of 120 PVC Sherman-lookalike traps were used across the study, resulting in 4800 trap nights for a total of 40 trapping days. Trapping sessions across the study were set for 10 consecutive days each month. This was conducted to enhance trapping success because the study took place in a small, protected area compared to similar studies in larger, protected areas that trapped for eight consecutive days [42,43].

#### 2.4.3. Mammal Identifications

All large mammal species identified in this study could be readily identified using the Stuart, 2013 [39] field guidebook. For data analysis, reconciled data after reaching a consensus was used as the official record of tracks recorded [44]. There was typically little variation between observers across the study.

For small mammal captures, the protocol of Mills et al. [21] was used to prepare the trap baits and set traps and was checked twice a day, once in the morning between 6 am and 10 am and in the afternoon between 5 pm and 6 pm. Traps were insulated when baiting with wet cotton wool to allow the mammal to survive the heat. Trapped individuals were identified at the species level [8]. The trapped animals were transferred from the traps to a clear plastic bag and weighed (to the nearest gram) using a German Calibration Service handheld spring balance. 

To recognise previously captured individuals, each captured individual was marked by trimming the hair on the back of the neck with a pair of scissors to reveal the undercoat colour, which varies [44,45]. Gentian violet was sprayed on the clipped area, a semi-permanent marking, to assist in identifying recaptures. However, during the study, there were no recaptures. This study was approved by the Reserve Management of the City of Tshwane (Protocol number: GNR 2019/21).

### 2.5. Statistical Analysis

#### 2.5.1. Large Mammals

Menhinick, Margalef’s Shannon–Weiner, Simpson, and Pilou indices were calculated as per Appendix A using R statistical software. Non-parametric tests were used for hypothesis testing, otherwise data were log-transformed to meet normality requirements. One-way ANOVA with log-transformed data was used to evaluate the differences between plots using the diversity indices and the abundance index at a 5% level of significance. In addition, ANOSIM was used to assess large mammal species abundances. Furthermore, the Simpson and Shannon–Wiener indices were used to measure the abundances of large mammal species across treatments [46]. In addition, Non-Metric Multidimensional Scaling (nMDS) calculations were constructed to gauge similarities between treatments.

A multiple regression analysis was applied to tests if species richness was influenced by the distance of treatment from the water source or *L. camara* density around treatment plots. The number of tracks was the outcome variable and the distance of tracks from the water source and the density of *L. camara* around the treatments were set as predictor variables. A Pearson’s product-moment correlation test was applied to ascertain the relationship between the density of *L. camara* around the treatments and the distance of treatments from the water source.

Lastly, a Generalised linear mixed model (GLMM) with a Poisson distribution using the *glmer* function in the lme4 package of the R statistical software package was used to assess whether either vegetation type or foraging guild affect species richness. The number of tracks identified inside the treatments was analysed as the response variable, whereas vegetation type, foraging guild, and their interactions were set as predictor variables.

#### 2.5.2. Small Mammals

Nonparametric species richness estimator Jack1 (i.e., first-order Jack-knife) and Chao1 [47,48] were used to detect changes in species richness. Three diversity indices were also calculated: Margalef’s index, the Simpson diversity index, and the Shannon–Wiener index of Diversity.

Data were tested for normality and—when necessary—transformed to be normally distributed. Then, a one-way ANOVA was used to determine how species differ in richness across treatments. A two-way ANOVA test was also applied to assess species diversity between the treatments using the Shannon index as the response variable with treatments and seasons set as independent variables.

Two separate GLMMs were used to assess relationships between Model (1), treatment type and the number of rodents captured, and Model (2), the weight of the rodents captured and season across treatments. The weight of the rodents and the number of rodents captured were analysed as outcome variables for Model 1 and Model 2, respectively, with the type of treatment and season as independent variables.

Each GLMM included the distance of treatments from the road as a random factor. The best model was checked by computing the AIC value because it is more robust for smaller sample sizes. Graphs for this study were produced using the GGplot2 package in the R statistical software package, and all data were analysed using this software.

## 3. Results

### 3.1. Large Mammals Results

A total of 829 individual large mammal species tracks comprising ten species from four foraging guilds were recorded by this study as follows: browsers (n = 287 tracks), grazers (*n* = 465), mixed feeders (*n* = 34), and carnivores (*n* = 43) across the 12 treatments and two vegetation types inside the GNR. Only *Equus quagga*, *Canis moesomelas*, and *Giraffa camelopardalis* were found across all the treatments; some individuals were detected in multiple plots within and/or across treatments and some individuals were detected across multiple days throughout the study. Large mammal tracks were recorded most in treatments that had never been invaded by *L. camara* (186 tracks). This number decreased with time from when invasions began, reaching 96 tracks in plots with 20 years of invasion. When sites were cleared, the species recovered, with 155 tracks in plots cleared in the period of 3–5 years (Appendix A).

#### 3.1.1. Differences between Treatments

For large mammal diversity indices, the ANOVA revealed no statistically significant difference between treatments (F_5_ = 0.233, *p* = 0.945). However, the highest diversity (Shannon and Simpson) and richness (Menhinick’s) indices were in the control treatment (Figure 1). In addition, the lowest species diversity was in the *ca*. 20 yrs treatment plots (Figure 1).

All ANOSIM tests for pairwise comparisons of treatments were significant (*p* = <0.05, Table 2). The strengths of the differences are shown in Table 3.

The nMDS results show differences in species composition across the treatments. The control treatments had the largest species composition (Figure 2).

#### 3.1.2. Treatments’ Proximity to the Water Source and *L. Camara* Density

For plots where large mammal tracks were sampled, regression indicated that the two independent variables (distance to the water source and *L. camara* density) explained 70.0% of the variance (R^2^ = 0.735, F_2,15_ = 20.89, *p* < 0.001). Distance of treatments from the water source was not a significant predictor of species richness (β = −0.07, *p* = 0.37; Figure 3), but the density of *L. camara* in the treatments was a significant predictor of species richness (β = −0.55, *p* = 0.036; Figure 3A). The relationship between the predictor variables, the density of *L. camara* inside the treatment plots, and the distance of treatments from the water source was not significant (t = 1.919; d = 16; *p* = 0.073). However, species richness was far higher in plots with very little *L. camara* (Figure 3A).

#### 3.1.3. Variation in Species Richness by Vegetation Type and Foraging Guilds

The GLMM results showed that the vegetation type (Wald χ^2^_2_ = 120.156; *p* < 0.01) and foraging guild of large mammals (Wald χ^2^_3_ = 76.771; *p* < 0.01) were significant predictors of large mammal densities. Overall, the highest number of large mammals was recorded in the bush-dominated sites compared to the grass-dominated sites. Furthermore, the study recorded, in order of numbers, grazers followed by browsers, carnivores, and mixed feeders, respectively (Figure 4). Interestingly, only grazers were found across both vegetation types, but the median of grazers in the grass-dominated plots was very negligible (~10) in comparison to the median in the bush-dominated plots (Figure 4). The highest number of grazers followed by browsers, carnivores, and mixed feeders, respectively, were found in the bush-dominated plots (Figure 4). Significant differences in large mammal densities were found for all treatment types except for *cl*. 20-year plots (Table 4), and a significant difference was found for Indigenous grass-dominated sites (Table 4).

### 3.2. Small Mammals Results

#### 3.2.1. Rodents Trapping

The study captured 66 individual rodent species of the Muridae family, which includes mice and rats. *Rhabdomys pumilio* (de Winton, 1897) [49,50] was the most captured species (*n* = 38; Appendix A) across the study. Because the two *Aethomys* species (*A. chrysophilus* and *A. ineptus*) occurring in South Africa cannot be distinguished from each other in the field [7], all *Aethomys* species captured by this study were analysed together as *Aethomys* sp. (Appendix A).

Trap success generally decreased with an increase in invasion by *L. camara*. The highest trap success was in the control area, the area that had not been invaded by *L. camara*, followed by the areas that had been cleared and those that had not been invaded in order of expected *L. camara* invasion (Figure 5). The area cleared in the previous 2 years had a lower trap success than the control but did not match the overall trend.

The Jack-knife 1 species richness estimators indicated that the study species inventory was between 80% and 98% complete, and the Chao1 showed it to be between 68% and 87% complete (Appendix A).

The ANOVA revealed a significant difference in rodent species richness across the treatments (F_5_ = 3.877, *p* = 0.025; Figure 1). Multiple pairwise comparisons were used to determine which treatment pairs were significantly different. The Tukey test revealed differences between the pairs *cl.* < 2 yrs–*ca*. 20 yrs, *cl*. 3–5 yrs–*ca*. 20 yrs and *ca*. 20 yrs–control (Figure 6), with adjusted *p*-values of 0.050, 0.020, and 0.001, respectively.

The results of a GLMM showed that treatment type (Wald χ^2^_5_ = 10.62; *p* = 0.050) was a significant predictor of the number of small mammals trapped, with the highest number of rodents trapped in the control followed by areas that had been cleared of *L. camara* for *cl.* 3–5 yrs, *cl* < 2 yrs, and areas that have been invaded by *L. camara* as follows: *ca*. 2 yrs, *ca.* 10 yrs, *ca.* 20 yrs, respectively (Figure 7).

There were significant differences in species richness found for treatments: *cl*. 3–5 yrs, *cl.* < 2 yrs, *ca*. 2 yrs, and the control. However, no significant differences were found for treatments *ca*. 20 yrs (Table 5).

#### 3.2.2. Species Diversity by Treatments and Season

The ANOVA revealed statistically significant differences for species diversity between treatments (F_5_ = 0.934; *p* < 0.001) and by season (F_1_ = 9.122 *p* = 0.003). For most treatments, species diversity was higher during the spring season except for the *ca*. 2 yrs and the control (Figure 8).

#### 3.2.3. Rodent Weight by Seasons across Treatments

A series of separate GLMMs were applied as follows: models with (1) no interaction, (2) one-way, (3) two-way interaction for the weight of the rodents as the outcome variable. The model with a two-way interaction produced the lowest AIC value (3614.1) in comparison to the other two models.

The weight of the rodents was significantly predicted by treatment (Wald χ^2^_5_ = 47.52; *p* < 0.001), season (Wald χ^2^_1_ = 16.98; *p* < 0.001) and species (Wald χ^2^_4_ = 129.53; *p* < 0.001; Figure 9). However, the two-way interaction between season and species (Wald χ^2^_4_ = 1.236; *p* = 0.873) did not predict the weight of the rodents. Rodents with the highest mean weights were found in the control area, in the spring season, and *Rhabdomys pumilio* was the most captured rodent across all treatments (Figure 9).

There were significant differences found for the weight of the rodents for treatments *ca*. 2 yrs, *cl*. 3–5 yrs; control for season: summer and for species: *Lemniscomys rosalia, Mastomys coucha*, and *Saccostomus campestris* (Table 6).

## 4. Discussion

The study assessed the impact of an invasive shrub *L. camara* on mammals of the Groenkloof Nature Reserve. For large mammals, the highest species abundance, richness, and diversity occurred where no *L. camara* had ever invaded. For small mammals, their weight was found to vary by treatment type, and these weights also varied by season across different treatments. The distance of treatments from the water source had no effect on large mammal species richness, which is especially surprising because large mammals are reported by most studies to frequent a water source to hydrate [51,52,53,54,55]. One explanation for this lack of effect could be that distances to the water are relatively short for a large mammal and do not affect long-term species richness. Nevertheless, the *L. camara* density around the treatments had a significant influence on species richness inside the GNR, as expected; treatments with the lower density of *L. camara* carried the highest species richness. A number of studies have reported that invasive plants can cause a significant impact on animals in the natural environment [5,15,51]. Pyšek et al. [56] reported that invasive plants caused a significant impact on resident animal richness in a global assessment study that included 167 invasive species belonging to 49 families. In South Africa, Dumalisile and Somers [15] published a study in 2017 that found that the invasive *Chromolena odorata* negatively affected large mammal species diversity in the Hluhluwe-Imfolozi Park. Studies have shown that landscapes with a higher density of invasive plant species can increase predation risk because they provide dense patches that can create an opportunity for carnivores to attack their prey [57,58].

Sampson et al. [59] reported that specific vegetation over general habitat types drives wildlife habitat use. In this study, the hypothesis that large mammal species richness would be affected differently across vegetation types based on their foraging guild was found to be true. Only grazers were recorded across the two vegetation types, but the highest number of grazers were recorded in the bush-dominated plots. *L. camara* has been shown to destroy grass where it occurs [5,60] and to limit or exclude native grass species through competition [61]. These effects are likely to explain the negligible number of grazers in the grass-dominated plots. Mixed feeders were only found in the bush-dominated plots, which could be due to similar reasons.

*Canis moesomelas*, the only carnivore found in the study, was affected by *L. camara* invasion across all feeding guilds. Dutra et al. [57] reported that invasive plant species alter consumer behaviour by providing refuge from predation. *C. moesomelas* were recorded in this study with potential prey such as the browser antelopes *Tragelaphus strepsiceros* and *Sylvicapra grimmia* and more in bush-dominated sites. The absence of mixed feeders and browsers in grass-dominated sites most likely limited the hunting range to bush-dominated sites. However, *C. moesomelas* are reported to be opportunistic and only rarely hunt antelopes when the availability of carcasses is reduced [62]. In a small, protected area like the GNR, this predation behaviour is more likely because there are no larger carnivores to kill prey for them to scavenge on. Thickets formed by *L. camara* could be a hindrance in this predation and could also provide a hideout for prey. These possibilities might explain the larger number of *C. moesomelas* found in the plots with no *L. camara.*

Most *Equus quagga* in the study were found in one *ca*. 2 yrs treatment that was next to a natural water source. This finding could be attributed to the possibility that *Equus quagga* could be less sensitive to food shortages, as suggested by long-term research in the Serengeti ecosystem and in the Kruger National Park [37].

In Kenya, *L. camara* was reported to replace indigenous vegetation, threatening the habitat of *Hippotragus niger* antelope [63]. This should be a concern for the management of the study area because *Hippotragus niger* antelope were recently introduced to the reserve.

Plumptre et al. [63] reported food supply, and availability of vegetation for herbivores and prey for carnivores, as one of the determining factors for large mammal abundance. This could also be attributed to the fact that large mammals tend to neglect any disturbed habitat [64]. Thus, as expected, the study found the invasive weed *L. camara* to be linked with decreases in large mammal abundance relative to the control treatments and cleared treatments. In addition, *L. camara* affected the space use of these large mammals. These findings are consistent with a global assessment study by Schirmer et al. [56].

For small mammals, the most captured animal was a rodent, *Rhabdomys Pumilio*, which was not surprising because this mouse is common and widely distributed in Southern Africa, especially in the urban Gauteng protected areas. Skinner and Chimimba [7] reported that it entered residential areas in several localities in the Gauteng Province. Julius et al. [65] captured 84 *Rhabdomys* sp. in one locality in Pretoria. The negligible number of *Aethomys* sp. in the study was also not surprising because a scan of the literature did not reveal any capture of this species in the Gauteng province. However, this species has been successfully captured in the Limpopo [66,67,68] and KwaZulu-Natal [69] provinces.

ANOVA results indicated that rodents’ species richness and diversity differed between the treatments. From the results, species richness increased for decreasing invasion. This finding is consistent with a study by Shackleton et al. [69] in Eastern Africa that showed the negative impacts of the invasive *C. odorata* on native biodiversity as far as species richness was concerned. A significant relationship was also found between trap success and the diversity indices. Overall, MacDonald [70] also showed that rodent richness and diversity decreased with increasing invasion in Western Ghana. Moreover, Baker and Frischknecht [71] reported a higher small mammal population on recently cleared juniper Rangeland in Utah compared to the invaded areas. These findings suggest that habitat preference decreases with an increase in the duration of weed invasion. One likely reason for this pattern is that, apart from some rodents such as the *Aethomys* spp., most rodent species are unable to tolerate the high (up to 90%) shade under *L. camara* thickets [72].

Treatment, season, species, and all the interactions between these variables significantly predicted the median weight of the small mammal species. This study predicted that the biggest rodents in terms of weight would be found in the control area. Bigger rodents in terms of weight were found in the control area, consistent with the study prediction, and these bigger rodents were captured during the spring season. These findings were expected because most weed thickets cover native seeds preferred by granivorous rodents [27]. The *L. camara* weed has been reported to be unpalatable and poisonous to some mammals [73,74,75]; *L. camara* is toxic and invasion leads to an unavailability of natural indigenous food. In addition, the study took place during a drought period when natural food was scarce inside the reserve, which could have exacerbated the situation. In addition, the breeding activities and abundance of food have been reported to influence the movements of rodents [76].

In another laboratory study, Pour et al. [76] confirm that *L. camara* shows a pro-toxic effect on mice. Finally, this study confirms the avoidance of *L. camara* patches by rodents due to toxicity in the natural setup. Moreover, the reproductive competition model showed that striped mice (*Rhabdomys pumilio*) were group-living during the spring season when they become most territorial and population density was high, supporting the ecological constraints model of Schradin and Pillay [77], which could explain the biggest *Rhabdomys pumilio* captured in the study during the spring.

Overall, for all the mammals studied, the presence of *L. camara* negatively affected species abundance, richness, and diversity and how these species were distributed across the invaded and cleared areas.

## 5. Conclusions

In conclusion, this study emphasises the importance of alien plant invasions coupled with other factors such as season in contributing to a loss in mammal diversity. Alien plants are already known to impact other aspects of biodiversity. Thus, these results compound the need to control alien plant invasions in natural areas. The study also found that *L. camara* presence is associated with decreased mammal abundance and composition in invaded areas, which could cause ecosystem imbalance resulting from changes in mammal species distributions because of *L. camara*.

## Figures and Tables

**Figure 1 biology-12-00296-f001:**
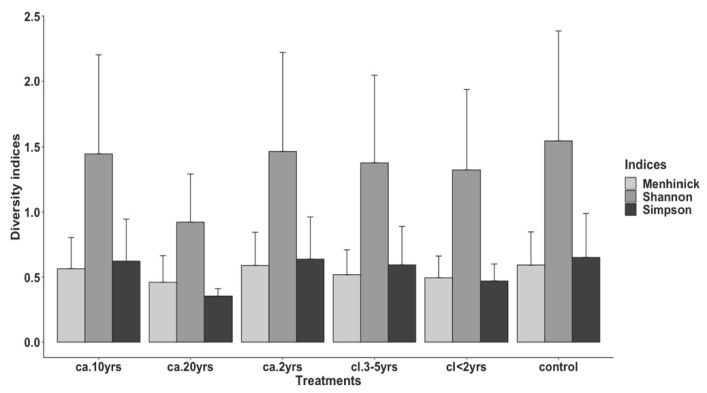
Mean ± SE of the diversity indices of large mammals of the Groenkloof Nature Reserve from areas of different treatments of *L. camara* invasion durations (*ca.* 2 yrs; *ca.* 10 yrs and *ca.* 20 yrs), differing *L. camara* clearing times (*cl* < 2 yrs and *cl.* 3–5 yrs), and a control area with no history of *L. camara* invasion. See Appendix A for treatments and indices definitions.

**Figure 2 biology-12-00296-f002:**
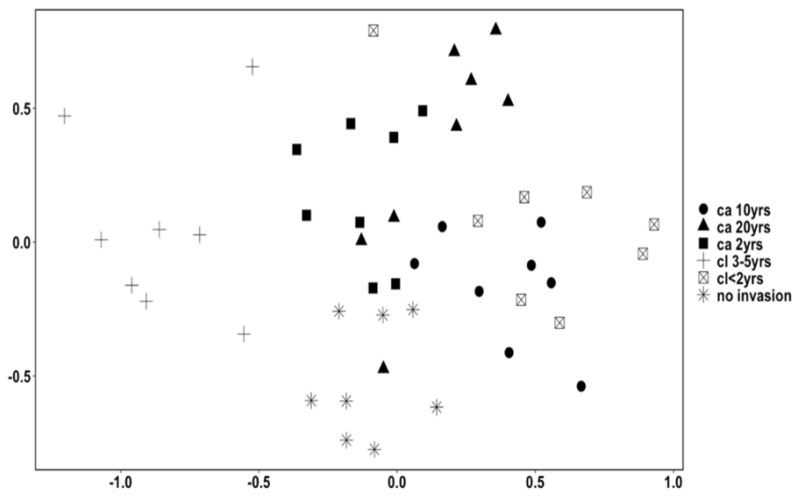
MDS plot (derived from Bray_Curtis dissimilarity in square root transformed data, Stress value = 0.148) showing large mammal assemblages of the six areas with differing *L. camara* invasion durations (*ca*), differing *L. camara* clearing (*cl*) times, and a control area with no *L. camara* invasion in Groenkloof Nature Reserve, South Africa.

**Figure 3 biology-12-00296-f003:**
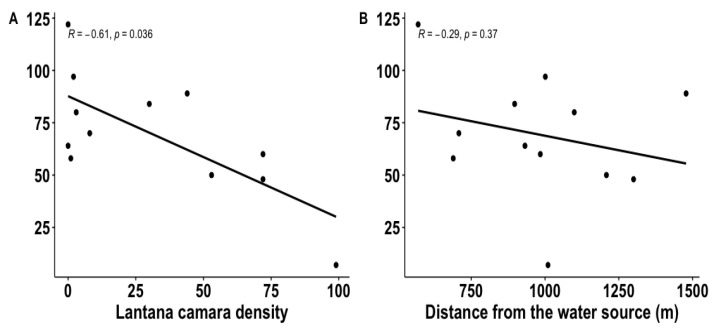
Scatterplots of the distribution of the overall species richness (track counts per species and per treatment type) by the density of *Lantana camara* around the treatments (**A**) and the distance of treatments from the water source (**B**) inside the Groenkloof Nature Reserve, South Africa.

**Figure 4 biology-12-00296-f004:**
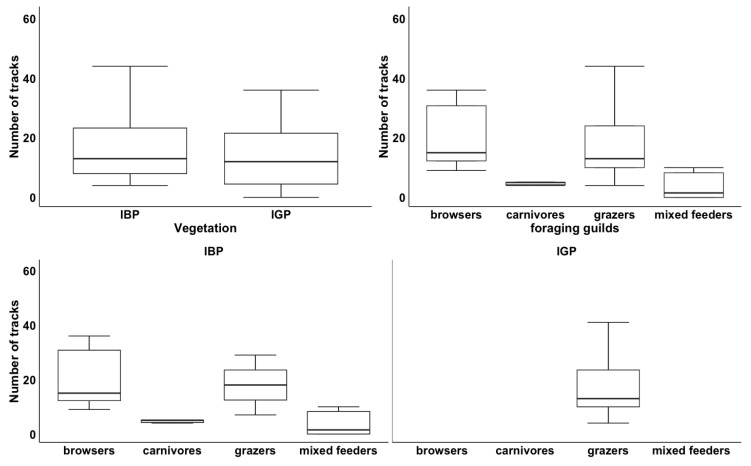
Frequency of occurrence of the number of tracks per treatment by vegetation type (top left), foraging guilds for both vegetation types (top right), and the interaction between vegetation and foraging guilds, bottom two plots (IBP = Indigenous bush-dominated plots, IGP = Indigenous grass-dominated plots). Boxes show medians (solid black line across the box) and 1st and 3rd quartiles.

**Figure 5 biology-12-00296-f005:**
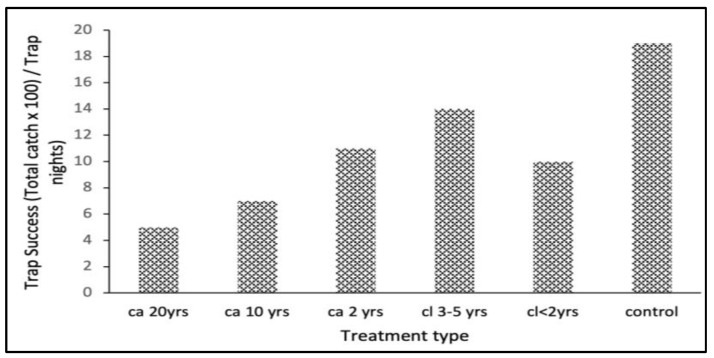
Trap success by treatment type inside the reserve.

**Figure 6 biology-12-00296-f006:**
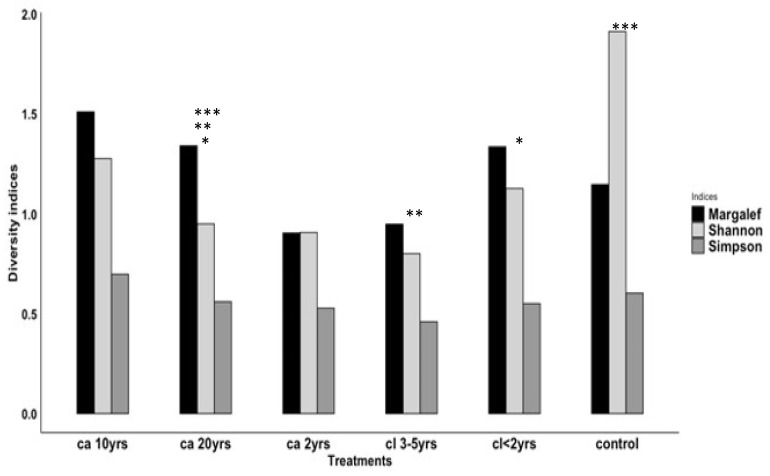
Showing the mean of the diversity indices for captured animals in six different treatments inside the reserve. Asterisks show the treatment pairs that are significantly different from each other.

**Figure 7 biology-12-00296-f007:**
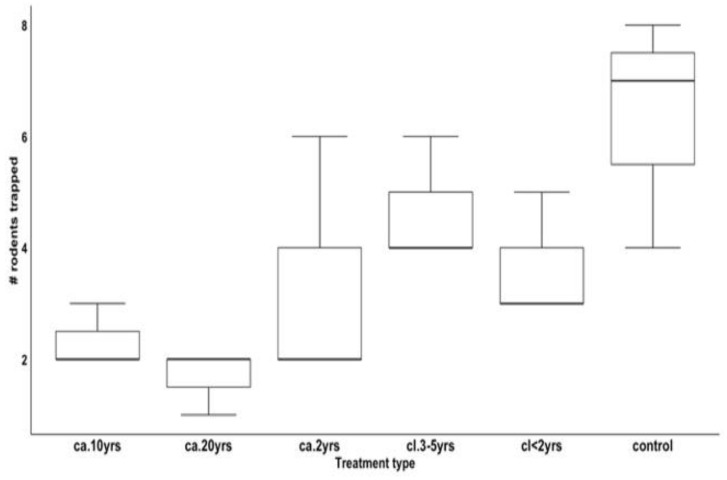
Counts of rodents trapped by treatment type.

**Figure 8 biology-12-00296-f008:**
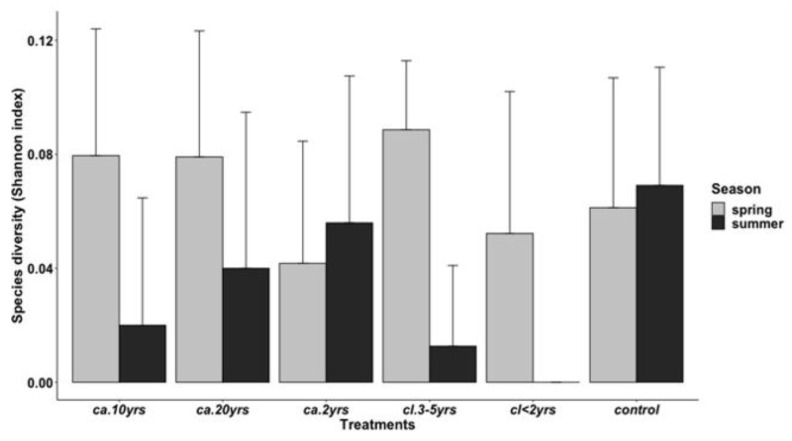
Mean ± SE of the species diversity of rodents of the Groenkloof Nature Reserve from areas of differing treatments of *Lantana camara* invasion durations (*ca.* 2 yrs, *ca.* 10 yrs, and *ca.* 20 yrs), differing *Lantana camara* clearing times (*cl*. < 2 yrs and *cl*. 3–5 yrs), and a control area with no history of *Lantana camara* across two seasons.

**Figure 9 biology-12-00296-f009:**
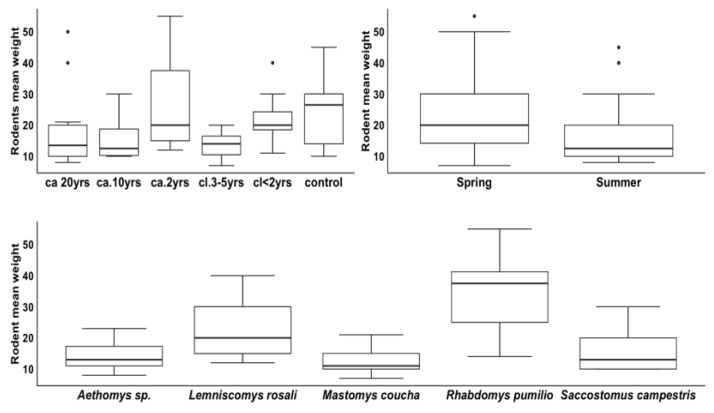
The median weight of rodents by treatments, season, and species inside the Groenkloof Nature Reserve. (*Lantana camara* invasion durations (*ca*), differing *Lantana camara* clearing (*cl*) times, and a control area without *Lantana camara* invasion inside the Groenkloof Nature Reserve, South Africa). Dots outside the boxes represent outliers.

**Table 1 biology-12-00296-t001:** Environmental conditions of the study area.

Environmental Factors	Values and SI Units
Mean annual temperature	18.7 °C–24.9 °C
Average Maximum Temperature recorded	24.5 °C–36 °C
Mean Annual precipitation	670 mm
Altitude	1082 m–1899 m above sea level

**Table 2 biology-12-00296-t002:** Simper pairwise comparison test between the six treatments inside the Groenkloof Nature Reserve, South Africa. Significant values are indicated by an asterisk.

Simper Pairwise Tests between Treatments	Test Statistics (R)	Significant level%
control—*ca*. 2 yrs	0.941	0.1 *
control—*ca*. 10 yrs	0.993	0.1 *
control—*ca*. 20 yrs	0.873	0.1 *
control—*cl*. < 2 yrs	1.000	0.1 *
control—*cl*. < 3–5 yrs	0.806	0.1 *
*ca.* 2 yrs–*ca.* 10 yrs	0.895	0.1 *
*ca.* 2 yrs–*ca.* 20 yrs	0.883	0.1 *
*ca.* 2 yrs–*cl.* < 2 yrs	0.998	0.1 *
*ca.* 2 yrs–*cl.* < 3–5 yrs	0.88	0.1 *
*ca.* 10 yrs–*ca*. 20 yrs	0.115	3.2 *
*ca.* 10 yrs–*cl.* < 2 yrs	0.996	0.1 *
*ca.* 10 yrs–*cl.* < 3–5 yrs	0.671	0.1 *
*ca.* 20 yrs–*cl.* < 2 yrs	0.849	0.1 *
*ca.* 20 yrs–*cl.* 3–5 yrs	0.531	0.1 *
*cl*. < 2 yrs–*cl.* 3–5 yrs	0.621	0.1 *

**Table 3 biology-12-00296-t003:** Pairwise ANOSIM comparisons for the different treatment plots. Significant values are indicated by an asterisk.

	Control	*ca*. 2 yrs	*ca.* 10 yrs	*ca*. 20 yrs	*cl.* < 2 yrs
*ca*. 2 yrs	0.211				
*ca*. 10 yrs	0.235	0.874			
*ca.* 20 yrs	0.065	0.393	0.186		
*cl.* < 2 yrs	0.019 *	0.034 *	0.051	0.187	
*cl*. 3–5 yrs	0.014 *	0.019 *	0.027 *	0.056	0.335

**Table 4 biology-12-00296-t004:** The output of a generalized linear mixed model fitted by a maximum likelihood (Laplace Approximation) showing treatment types and vegetation type. Significant values are indicated by an asterisk.

Variables	Estimate	Std Error	Z Value	*p* Value
Treatments—*ca.* 20 yrs	0.040	0.057	0.709	*p* = 0.478
Treatments—*ca.* 2 yrs	2.967	0.355	8.341	*p* < 0.001 *
Treatments—*cl*. 3–5 yrs	5.760	0.278	7.090	*p* < 0.001 *
Treatments—*cl.* < 2 yrs	7.078	0.954	74.163	*p* < 0.001 *
Treatments—control	6.552	0.139	46.848	*p* < 0.001 *
Vegetation–IGP	0.208	0.997	2.134	*p* = 0.032 *

**Table 5 biology-12-00296-t005:** The output of a GLMM model showing species richness by treatment types. Significant values are indicated by an asterisk.

Parameters	Estimates	Std Error	Z Value	*p* Value
treatments—*ca*. 20 yrs	1.099	5.774 × 10^−1^	1.903	*p* = 0.061
treatments—*ca*. 2 yrs	−2.340 × 10^1^	6.965 × 10^4^	0.000	*p* = 0.040 *
treatments—*cl*. 3–5 yrs	2.877 × 10^−1^	7.638 × 10^−1^	0.377	*p* < 0.001 *
treatments—*cl.* < 2 yrs	−4.055 × 10^−1^	9.129 × 10 ^−1^	−0.444	*p* = 0.001 *
treatments—control	−1.099	1.155	−0.951	*p* = 0.023 *

**Table 6 biology-12-00296-t006:** Output of a GLMM model for comparisons of the weight of the rodents by treatments, seasons, and species. Significant predictors are shown with asterisks.

Variables	Estimate	Std Error	Z Value	*p* Value
Treatments: *ca.* 10 yrs	−0.211	0.113	−1.855	*p* = 0.063
Treatments: *ca.* 2 yrs	0.356	0.112	3.171	*p* = 0.001 *
Treatments: *cl.* 3–5 yrs	−0.275	0.135	−2.031	*p* = 0.042 *
Treatments: *cl.* < 2 yrs	0.188	0.117	1.600	*p* = 0.109
Treatments: control	0.309	0.118	2.619	*p* = 0.008 *
Season: summer	−0.356	0.161	−2.205	*p* = 0.027 *
Species: *Lemniscomys rosalia*	0.390	0.135	2.873	*p* = 0.004 *
Species: *Mastomys coucha*	−0.142	0.150	−0.947	*p* = 0.343 *
Species: *Rhabdomys pumilio*	0.845	0.127	6.617	*p* < 0.001
Species: *Saccostomus campestris*	0.042	0.146	0.293	*p* = 0.769 *
Season: Summer—Species: *Lemniscomys rosali*	0.172	0.201	0.855	*p* = 0.392
Season: Summer—Species: *Mastomys coucha*	0.026	0.236	0.114	*p* = 0.909
Season: Summer—Species: *Rhabdomys pumilio*	0.159	0.188	0.843	*p* = 0.398
Season: Summer—Species: *Saccostomus campestris*	0.098	0.218	0.451	*p* = 0.652

## Data Availability

The data set used for this study is with the corresponding author and is available upon request.

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
