# Peer review of "Effects of the Density of Invasive Lantana camara Plants on the Biodiversity of Large and Small Mammals in the Groenkloof Nature Reserve (GNR) in South Africa"

_biology, 2023, doi:10.3390/biology12020296_

Round 1

Reviewer 1 Report

Dear Author

The following comments should be taken into consideration: 

-Abstract lines 28-36: It would be helpful if you provided more statistical results and findings in the abstract.

-Introduction: The main problems and objectives of your study should be clearly stated. 

-Line 51: Present as Ewel et al. [1999] identified... This format should be considered throughout the entire text.

-Lines 60-69: Sentences with little specificity. Rewrite needed.

-Lines 116-121: Removed. Not related to materials and methods!

-Lines 147-175: The heading does not relate to this. Make sure the headings are appropriate!

-Line 206 (Data Collection): This section should be summarized. Unnecessary data have been included.

-Line 242 (Statistical analysis): The statistical analysis was explained in 61 lines! This section should be summarized.

-Lines 326-327: Replace with this sentence: "When diversity indices were used, the Anova revealed no statistically significant difference between treatments"

-Figure 2: Provide more quality version of this figure.

-Lines 372-373: Make sure the values are correct!

Author Response

See attached response file, The corrections are highlighted in yellow and addressed in the manuscript

Reviewer 2 Report

Reviewer comments

Journal: Biology (ISSN 2079-7737) 

Manuscript ID: biology-2164666

Title: “The influence of weed on African mammals". 

By reviewing the entire attached manuscript, I found that the manuscript idea is good, the topic fits well with the scope of Biology -MDPI, and this point is interesting but it lacks novelty, as some authors have worked on points close to it.

However, the scientific writing is good but needs more revisions to improve it. The figures are also good, and the tables are also clear, but they lack the most important thing, which is the absence of stars or letters that shows the significance between the means. However, this manuscript will be deserved a major revision before consideration for publication in Biology -MDPI.

Please find the comments in the attached file.

Author Response

Please see the attached revised manuscript responses for reviewer 1 and 2 comment.

And below is the response for Reviewer 2 comments.

Comment 1 response: The study novelty is highlighted in the revised manuscript. See Lines 55-79.

Comment 2 response: The manuscript has been revised extensively based on the comments from Reviewer 1 and 2 

Comment 3 response: Asterisks has been included in figure 5 to show significance between the means. See Figure 5.

Round 2

Reviewer 2 Report

Reviewer comments-R2

Journal: Biology (ISSN 2079-7737) 

Manuscript ID: biology-2164666

Title: “The influence of weed on African mammals". 

By reviewing the revised version of the latest manuscript, I found that the authors did not respond to many of my previous comments, and they did not even care about responding to them. The simplest of those comments that I asked to respond to are that they follow the instructions of the Journal regarding writing references, the names of scholars, and other previous comments on the entire manuscript.

Therefore, this manuscript still needs a major review before it is accepted for publication in  Biology -MDPI or resubmit  to the journal's website after a full correction has been made.

Please find the comments in the attached file, which was sent previously.

Author Response

Thank you for all the constructive comments  and we believe to have adressed all of them to the best of our ability with the responses to your comments highlighted as yellow on the attached document

Round 3

Reviewer 2 Report

I agree to publish after a comprehensive linguistic review of the entire manuscript

Author Response

The manuscript has now been professionally edited. Also please find attached the newly revised figures to replace the older ones.